# A Review: Late Wilt of Maize—The Pathogen, the Disease, Current Status, and Future Perspective

**DOI:** 10.3390/jof7110989

**Published:** 2021-11-19

**Authors:** Ofir Degani

**Affiliations:** 1Plant Sciences Department, Migal-Galilee Research Institute, Tarshish 2, Kiryat Shmona 11016, Israel; d-ofir@migal.org.il; 2Faculty of Sciences, Tel-Hai College, Upper Galilee, Tel-Hai 12210, Israel

**Keywords:** *Cephalosporium maydis*, chemical control, crop protection, disease cycle, fungus, *Harpophora maydis*, *Magnaporthiopsis maydis*, real-time PCR

## Abstract

Late wilt (LWD) is a vascular wilt disease that outbursts late in maize development, usually during or after flowering. The disease causal agent, the soil and seed-borne fungi, *Magnaporthiopsis maydis*, causes significant economic losses in Egypt, Israel, Spain, Portugal, and India. Since its discovery in the early 1960s in Egypt, the knowledge base of the disease was significantly expanded. This includes basic information on the pathogen and its mode of action, disease symptoms and damages, methods to study and monitor the pathogen, and above all, control strategies to restrain *M. maydis* and reduce its impact on commercial maize production. Three approaches stand out from the various control methods inspected. First, the traditional use of chemical pesticides was investigated extensively. This approach gained attention when, in 2018–2020, a feasible and economical treatment based on Azoxystrobin (alone or in combination with other fungicides) was proven to be effective even in severe cases of LWD. Second, the growing trend of replacing chemical treatments with eco-friendly biological and other green protocols has become increasingly important in recent years and has already made significant achievements. Last but not least, today’s leading strategy to cope with LWD is to rely on resistant maize genotypes. The past two decades’ introduction of molecular-based diagnostic methods to track and identify the pathogen marked significant progress in this global effort. Still, worldwide research efforts are progressing relatively slowly since the disease is considered exotic and unfamiliar in most parts of the world. The current review summarizes the accumulated knowledge on LWD, its causal agent, and the disease implications. An additional important aspect that will be addressed is a future perspective on risks and knowledge gaps.

## 1. Introduction

Maize (corn, *Zea mays* L.) is considered one of the world’s most important human food supplies and the third leading crop, after wheat and rice [1]. It is grown by most countries for human food, animal feed, and other uses. The United States and China supply half of the international market [2]. The estimated average economic loss due to reduced yield caused by maize diseases in the United States and Ontario, Canada, during the years 2016–2019 was $US138.13 per hectare [3].

The maize late wilt disease (LWD) was first identified and reported in Egypt in 1961–1962 [4,5] and gradually reported in other countries. The disease is considered the most severe threat to commercial maize production in Israel [6] and Egypt [7], and a major concern in other maize-growing countries such as India [8], Spain, and Portugal [9]. Late wilt was not reported in the United States. Nevertheless, this disease is considered a threat to maize production in this country [10], and a Plant Protection and Quarantine [11] emergency plan for LWD is updated periodically. Direct economic impact attributable to LWD in new countries is difficult to predict. Besides yield losses, it may have severe effects since it could lead to restricted agricultural products and equipment movement, long-term quarantine, and crop embargo.

While research on the LWD was led by Egyptian groups in the first decade since its discovery, it attracted researchers from different countries, and the knowledge base expanded significantly. As expected, many of these studies were directed towards examining a range of agrotechnical, chemical, and biological control strategies to cope with this emerging risk. The past two decades’ introduction of molecular-based diagnostic methods to track and identify the pathogen (which started in 1999 [12]) enhanced this global effort. Marked progress in our understanding of the disease’s mode (the pathogenesis) and its causal agent, the fungus *Magnaporthiopsis maydis*, was achieved. Still, global LWD research efforts are restricted since the disease is considered exotic and unfamiliar in most parts of the world. 

The variety of tools developed to study and monitor *M. maydis* were recently reviewed by us [13] and will be summarized here briefly. Also, reviewing the global efforts to develop LWD control methods is worthy of a separate paper and will be presented here in summary. The current review summarizes the knowledge accumulated on LWD, its causal agent, pathogen spread, disease implications, and control methods. Another essential aspect discussed is a future perspective on risks and knowledge gaps that should be addressed. 

## 2. The Pathogen

*Magnaporthiopsis maydis* reproduce asexually through sclerotia and spores, and so far, no perfect stage has been identified [14]. This pathogen can be considered necrotrophic since it thrives on the remaining dead tissues [15] after killing the maize host [4]. However, it survives for an extended period (up to 80 days) on living susceptible maize plants and asymptomatically in resistant maize cultivars and alternative hosts (as will be detailed in Section 3). Thus, it may be defined more accurately as hemibiotrophic. The taxonomic position of the late wilt causal agent has shifted over the years. The fungus’ most common name in recent years is *Magnaporthiopsis maydis* (Samra, Sabet, and Hing; Klaubauf, Lebrun, and Crou); additional synonyms are *Harpophora maydis* and *Cephalosporium maydis* (Samra, Sabet, and Hingorani).

Another synonym of the fungus, mentioned in several references, is *Acremonium maydis*, the black bundle disease agent. This is probably an identification mistake. According to Sabet et al. 1970 [15] and the Compendium of Corn Diseases (APS Press, St. Paul, MN, USA, 2016) [16], *Acremonium maydis* is a different fungus. Indeed, Samra et al. [17] could differentiate between the late wilt fungus of maize (*M. maydis*) and other related *Cephalosporium* spp., including *C. acremonium*, based on microscopic characteristics (conidia, conidiophores, and hyphae morphology), growth conditions, and pathogenicity in maize. Since the Egyptian, Indian and Hungarian isolates of *M. maydis* differ in morphology, pathogenicity, and route of infection [18], the fungus identity confirmation is complex. It should rely on molecular targeting of the fungus species-specific fragment [19], as discussed at the end of Section 4.1 and in Section 5.2.

Gams (2000) presented the genus *Harpophora* based on *H. radiciola* for a group of species that are phialophora-like in morphology [20]. Ward and Bateman (1999) [12] and Yuan et al. (2010) [21] generated universal internal transcribed spacer (ITS) phylogeny in which the species of *Harpophora* were close to or grouped with *Gaeumannomyces* Arx & D.L. Olivier. To strengthen this, Saleh and Leslie (2004) [19], based on ITS, b-tubulin, histone H3 gene, and *MAT-2* sequences, transferred the fungus to the *Gaeumannomyces-Harpophora* species complex. Based on amplified fragment length polymorphism (AFLP) profiles, they conclude that the late wilt pathogen is distinct from the other tested species of *Cephalosporium, Phialophora* sensu lato and members of *Gaeumannomyces*-*Harpophora* species complex. Thus they classified it as *Harpophora maydis.* A decade later, a two-locus (LSU, RPB1) phylogeny was carried out by Klaubauf et al. (2014) [22], enabling the repositioning of *Magnaporthiopsis maydis* to the genus *Magnaporthiopsis* [23].

## 3. Primary and Alternative Host Plants

Maize is considered the primary host of *M. maydis*. The pathogen can also cause a significant damping-off and stunting of *Lupinus termis* L. (lupine, Albus cv.), which is widely cultivated in Egypt [24]. In addition, it was reported that *M. maydis* infected *Gossypium hirsutum* L. (cotton) [25]. The cotton Bahteem 185 cv. showed increased production of root laterals, and local dark red lesions and shallow cracks appeared on cotton sprouts’ roots (up to 45 days from sowing, DAS). Later, as the plants mature, these lesions disappear [25].

More recently (2019), a field survey and growth chamber pathogenicity trials accompanied by a quantitative real-time PCR (qPCR) analysis confirmed *M. maydis* target DNA present in the roots of cotton (Pima cv.) plants grown in infested soil [26]. The pathogen infection damaged 37 old cotton sprouts’ root biomasses. Yet, tracking the effect of *M. maydis* soil inoculation on cotton plants under field conditions throughout the whole season in a two-year study shows that the fungus did not affect cotton growth parameters or yield [27]. These results imply that the maize pathogen maintained an endophytic lifestyle in cotton plants. Yet, under drought stress, *M. maydis* infection led to decreased growth of the cotton plants (without any measurable effect on yield production) [27]. So, some opportunistic behavior of this pathogen in cotton could exist.

*Magnaporthiopsis maydis* DNA was also identified in sprouts of *Citrullus lanatus* (watermelon, Malali cv.) and the grass *Setaria viridis* (green foxtail, green millet) [26]. Infected watermelon (up to 37 DAS) had delayed emergence and development, were shorter, and had reduced root and shoot biomass. In *S. viridis,* the fungal infection rate (infected plants percentage) and severity (*M. maydis* DNA relative levels) were higher than maize. Nonetheless, there was no significant effect on *S. viridis* growth or development, and the plants did not develop disease symptoms.

The *S. viridis* results are particularly interesting. This grass species is a commercial crop in Israel but is also a wild native species in Eurasia, including Israel. Moreover, it is known on most continents as an introduced species and is closely related to *Setaria faberi*, a noxious weed [28]. The presence of *M. maydis* in this hardy soil, which grows in many types of habitats, reveals a worrisome risk of the pathogen’s spread.

The limited screen to identify alternative host plants for *M. maydis* encourages carrying out a more in-depth study. Such a study should scan other potential host plants and examine the unique relationship of this fungus with each of them. In particular, this effort should focus on hosts that are cultivated alternately to maize in a crop rotation.

## 4. *Magnaporthiopsis maydis* Distribution

### 4.1. Global Distribution

Since its first discovery in Egypt in the early 1960s, LWD was gradually reported in other countries in what seems like an expanding geographic distribution of this fungus. Late wilt was reported in Egypt (1961) [5], India (1970) [29], Hungary (1998) [30], Spain and Portugal (2011) [31], Israel (2018) [32] and possibly in Nepal (2015) [33]. In Nepal, the report referred to *Cephalosporium acremonium*, a synonym of the black bundle disease agent [34,35]. Still, this pattern may be the consequence of increasing recognition of the pathogen. The identification and report of the disease may be made many years after its first appearance, as it happened in Israel. In this case, maize plants with typical wilt symptoms were observed since the 1980s, and the pathogen was identified in 2002 (A. Sharon, personal communication) and reported in 2013 [32]. Unconfirmed reports summarized by Johal et al. (2004) [36] stated that LWD was discovered in Kenya, Romania and Italy. So, if we include the unconfirmed reports, LWD global distribution consists today of 10 countries (Figure 1). Yet, this number could be larger.

Factors involved in the global spreading of *M. maydis* include infested plant materials (especially seeds) and environmental changes. The pathogen can be dispersed through the movement of crop residues and soils. Cultivation equipment can spread the fungus between areas, and land tillage practices can distribute it within the field. Since the fungus is seed-borne [37], it was assumed that it reached Hungary by importing infected propagation stock. After it first appeared, the outburst and activity of the pathogen in Hungary was attributed to dry early summers and global warming [30]. In line with this hypothesis, it was found that the pathogen also spread in relatively resistant plants that showed no symptoms [32] and that seeds of these healthy, relatively resistant plants may therefore also spread the disease.

Identification of the wilting maize plant causal agent is challenging, and a correct diagnosis may be difficult *Magnaporthiopsis maydis* can be diagnosed based on morphological characteristics, but the process requires considerable time and taxonomic expertise. Recovery of *M. maydis* is problematic, even from heavily infested material, as noted by Saleh et al. [38]. This could be attributed to the pathogen’s slow growth and the relative abundance of other, more rapidly growing fungi, particularly *Fusarium* spp.

In Israel, during the 1990s, it was presumed that the maize plants wilting in commercial fields was the consequence of *Fusarium verticillioides* infestation since this pathogen was the most abundant in the infested plant samples [13]. Koch’s postulates and molecular assay eventually proved that the primary cause of the dehydration symptoms is *M. maydis*, whereas *F. verticillioides* is a secondary invader or opportunist that can develop in these attenuated maize plants [39].

Dehydration symptoms can result from insufficient water supply and are well-documented symptoms of other more familiar diseases. Stem symptoms may be worsened by secondary invaders such as *F. verticillioides* causing stalk rot and *Macrophomina phaseolina* causing charcoal rot [40,41]. The discovery of *M. maydis* 200 bp AFLP-derived species-specific fragment [19] and its use in PCR or qPCR [32,42] enables distinguishing *M. maydis* from other species rapidly and reliably. Similarly, these methods can be used for targeting the fungus ribosomal internal transcribed spacer (ITS) region [43,44].

### 4.2. Local Distribution

*Magnaporthiopsis maydis* is scattered in small quantities in the soil [13]. This scattered dispersal pattern may develop into disease patches in the field, seen as brown areas in an aerial photograph (Figure 2). The disease dispersal in the field can be affected by the cultivation (for example, sowing direction). Since low water potential is considered one of the most influential factors enhancing late wilt disease progression (to be discussed in detail in Section 5.4), ununiform irrigation, such as blockage sprinklers in the irrigation system or limited water supply at the edge of the field, evoke the late wilt disease burst (Figure 3).

Studies on *M. maydis* populations were carried out in Egypt [18], southern Portugal and Spain [9]. In Egypt, the disease spread rapidly since its discovery in 1961 until 1979 to most maize-growing areas [45]. Up to 100% of infected plants were reported in some farms. It was shown that *M. maydis* isolates differ in pathogenicity, morphology, and route of infection [18]. Most importantly, it was discovered in Egypt and Spain that this pathogen could undergo pathogenic changes resulting in highly virulent strains [9,14]. Variations in pathogenicity between *M. maydis* isolates were also reported in Israel [13]. The *M. maydis* Egyptian isolates were classified into four clonal lineages, revealing diversity in virulence and colonization ability on maize and competitiveness relationships [14]. The most virulent lineage (once tested alone) was the least competitive on susceptible maize when tested in a mixed inoculum of all four fungal lines. In contrast, one less virulent lineage dominated 70% of infections and appeared to be the most competitive.

Thus, it was hypothesized [14] that the highly virulent strain may have an advantage in this interspecies competition when the field is seeded with a resistant maize cultivar. Nevertheless, in areas where susceptible maize hybrids are cultivated, the less virulent *M. maydis* strains might become more abundant, and the most virulent lineage may be relatively rare. This may be one reason why extensive cultivation of one resistant genotype for several years may eventually result in the development of virulent or aggressive *M. maydis* strains [9,46] that can cause disease to that particular cultivar. This scenario has happened in Israel to the relatively resistant maize cultivar, Royalty, which became the leading sweet maize cultivar during the late wilt disease outbreak in the 1990s [13,47].

Other possible explanations should be explored. It should be taken into consideration that environmental aspects (biological and physical) are involved. *M. maydis* strains may be differentiated by their ability to grow either in the soil or the host plant, and their sensitivity to other soil microbes or their metabolic products. Such interactions will be discussed in detail in Section 6.3.

### 4.3. Magnaporthiopsis maydis Survival

In addition to being seed-borne, *M. maydis* can remain viable as vegetative dispersal bodies (sclerotia and spores) in soils or on maize residues for long periods. Usually, spores are produced after two days in a rich medium [17] and are at their highest abundance 4–6 days after incubation [15,48]. On days 3 and 9 of the cultures’ growth, they were almost devoid of spores. Sabet et al. [15] reported that *M. maydis* survival is restricted to the top 20 cm of the ground and that under field conditions, it is exacted nutritionally, adapts to host remains (maize straw), and is less able to decompose fresh substrate (supplemented wheat bran). The pathogen can persist on maize stubble for 12–15 months [35,49]. Sclerotia are produced under low humidity and ensure the long-term survival of *M. maydis* (up to 15 months) in no-tillage cultivation residues on the soil surface. In India, Singh and Siradhana (1987) [49] reported that *M. maydis* could survive for up to two months in seeds stored in the laboratory and 10 months in seeds stored at high temperature and low humidity. They also predicted more prolonged survival at low temperatures. Secondary hosts can efficiently support the pathogen’s survival in the long term (see Section 3). For example, it was shown that the pathogen’s DNA levels in the grass *S. viridis* are similar and even higher than those in the susceptible maize hybrid prelude cv. [50].

## 5. Pathogen Development and Pathogenesis

### 5.1. The Late Wilt Disease Cycle

The late wilt disease cycle is well documented (Figure 4). The molecular-based ability to monitor the pathogen DNA inside the host plant’s roots, stem leaves, and cobs during pathogenesis supported the results obtained by traditional methods [17,29,30,35,37]. According to Sabet et al. [35], the infection occurs during the first three weeks of growth. The pathogen easily infects maize sprouts, but as the plants grow, fewer are infected and none after ca. 50 DAS. The fungus penetrated the roots and was first identified in the xylem 21 DAS. On day 35 after sowing, the pathogen reaches the stem’s first internode. At 49 days, *M. maydis* spread to the fourth internode [35]. The pathogen is restricted in its ability to directly penetrate seedlings’ leaves, even if they are detached and wounded [6], although the leaves are probably a more accessible nutritional source. In root powder-supplemented medium, laccase levels were higher, and the fresh fungal weight was lower than in the same medium based on leaves’ powder [51].

At that stage (50 DAS), relatively low but identifiable amounts of fungal DNA were measured by PCR in various plant parts [32]. When tassels first emerged (day 63) [35], the fungus was found throughout the length of the stalk, although there was less of it toward the plant’s upper parts. The pathogen DNA levels peaked in the stems at this plant age, and the first disease symptoms appeared shortly after. The first symptom is moderate plant wilting [17] progressing upwards. The leaves change their color to light-silver and yellow-brown, and roll inward from the edges [35,41]. At the last stage of the disease near the harvest (75–85 DAS in sweet cultivars), the fungus can be traditionally isolated from the cobs [35] or more sensitively identified using PCR [32].

*Magnaporthiopsis maydis* was detected in different ear parts using isolation on potato dextrose yeast extract agar (PDYA) medium in Petri dishes from naturally infected mature maize plants [37]. These include ear branches, cob, seeds, ear husks, and silk. The pathogen was mainly established in the branch, followed by cob, seeds, husks, and silk. *Magnaporthiopsis maydis* is borne in seeds externally (in the seed coat) and internally (in the embryo and the endosperm) [37]. In the relatively resistant Amon cv. (yellow maize), the fungus was not detected by both agar and blotter standard methods and only proved to be established in the seed coat.

Here, the advantages of the molecular method stand out. The genotype Royalty cv. usually displayed no LWD symptoms and grew well, even in heavily infested field soils. Nevertheless, in this relatively resistant cultivar, fairly minor but constant and clearly identifiable pathogen DNA levels were measured by PCR throughout the growth period in all of the plants’ parts, including the seeds [32]. Interestingly, in a field trial, ca. two weeks after the regular harvest stage (87 DAF, 37 DAF), the Royalty cv. plants dried out, implying that the relative resistance of this cultivar is only temporary, and eventually, this cultivar will collapse [13].

### 5.2. Molecular Monitoring of the Pathogen during the Disease Stages

Tracking *M. maydis* DNA variations using the qPCR method [42] significantly improves our detection sensitivity and understanding of the pathogenesis. This method is far more sensitive than the traditional PCR method. It can detect variation in proportionate *M. maydis* specific DNA abundance normalized to the housekeeping cytochrome c oxidase (*COX*) gene DNA between 10 to 10^−5^ (a million times difference) [42]. Intriguing results were obtained by examining changes in *M. maydis* DNA in healthy and diseased fields during the spring and summer of 2018 [41]. In both fields, the pathogen’s DNA levels in the stem at 53–58 DAS were higher than those measured in the roots at 30–31 DAS. However, this trend changed in the healthy field spring plants, and *M. maydis* DNA levels dropped at the season end (73 DAS). In contrast, in the severely diseased summer field, the pathogen’s DNA levels intensified towards the season end (71 DAS). It can be assumed that these DNA variations are related to plant health and its immunity to the pathogen.

The weakening of the plants in severe late wilt cases may lead to sharp fungal DNA levels elevation within the host stem tissue [41,52]. However, when the plants show extreme dehydration, the pathogen’s DNA levels may drop [42]. We hypothesized that in such cases, the fungus enters into the asexual reproduction stage and develops spores and sclerotial bodies [36], while the primary hyphae biomass gradually comes apart. To support this conclusion, comparing the summer 2018 field experiment results to the summer 2017 results (conducted at the same site) published earlier [52] indicates that in both cases of severe disease outbreak, a sharp elevation in fungal DNA inside the stem tissue was measured (Figure 5). While that has been said, it should be emphasized that disease severity and yield loss cannot be inferred from the presence or amount of DNA in tissues. Such correlation should be examined and established in follow-up studies.

### 5.3. Effects of Abiotic Factors on the Biology of Magnaporthiopsis maydis

The use of poor water quality can result in various soil and cropping problems, including salinity (osmotic and ionic stress). Such stresses can also affect the LWD. The pathogen’s spore germination and hyphal development react differently to various ambient conditions in vitro [48]. *Magnaporthiopsis*
*maydis* spores were much more sensitive to oxidative stress (hydrogen peroxide or the superoxide-generating agent menadione) than the hypha that grew at a mild rate in these oxidative agents. In contrast, the pathogen’s colonies development was drastically reduced in a salty environment (hyperosmotic and ionic pressure), while the spore gemination was less affected. We hypothesize that the different locations in which those two developmental stages occur play a pivotal role in this outcome (Figure 6). Conidia are produced within the maize vascular vessels (the xylem) 35 days after sowing [35]. They enable the pathogen to survive for long periods outside the host’s internal environment under hostile conditions. They are germinated under favorable conditions (most important of which are moisture and temperature), and infected maize sprouts through the roots. So, spore germination often occurs in the ground, whereas salt pressure is expected. This probably led to the adaptation of the spore germination to such an environment.

In contrast, most hyphal developments occur within the host’s homeostasis inner surrounding in which salt pressure does not exist, but oxidative stress does. As a result, their growth is sensitive to the unfamiliar hyperosmotic and ionic changes while they develop immunity to oxidative pressure. Reactive oxygen species (ROS), either hydrogen peroxide (H_2_O_2_), the hydroxyl radical (^·^OH), or superoxide anions (O_2_^−^), are generated endogenously in many plant cells as a consequence of metabolic processes such as respiration, and also by the immune system in response to pathogens [53]. The late wilt pathogen invasion and spreading inside the plants probably evoke the plant’s hypersensitive defense reaction and the induction of oxidative stress. Such stress may also serve the pathogen’s dispersal timing, maintaining the spores at an ungerminated stage until they release from the host.

### 5.4. Environmental Conditions and Disease Development

During growth on maize leaves, *M. maydis* must overcome various kinds of physical and chemical ambient challenges such as temperature, light, pH, oxidative, osmotic, and other stresses. Therefore, investigating the effects of these factors on spore germination and mycelial growth is essential for understanding the pathogen and its relationship with the host plant, disease outbursts, and severity. Indeed, environmental stress tolerance and disease resistance may be linked to each other. For example, the maize cultivar Giza 2 (G2) was more resistant to late wilt disease and more tolerant to water stress and a combination of both factors than Population 45 (P45) cv. [54].

Optimal moisture and temperature conditions for maize growth may allow the disease development but are not correlated to *M. maydis* favorable conditions, especially its low moisture requirement [55]. The disease progresses rapidly at 20–32 °C, with optimum disease development at 28–30 °C [56]. The growth of *M. maydis* in the soil is repressed sharply above 35–36 °C and is low at 8–12 °C [17,30]. The favorable pH values for *M. maydis* growth are 5–6, although the fungus can adjust to a wide range of pH from 4.5–10 [48,49].

To expand our understanding of the environmental role in maize LWD, we compared the meteorological data, the symptoms, and the pathogen’s DNA levels inside the host in northern Israel (Hula Valley, Upper Galilee) in the maize growing seasons of 2016–2020 (Table 1). This comparison is reasonable since data were collected from nearby fields located ca. 10 km from each other. Also, the same maize genotype (Prelude, LWD-susceptible sweet maize) was inspected. Finally, the same valuation methods were applied (the cobs’ spathes dehydration symptoms’ evaluation and the lower stem section qPCR analysis). This summarization supports the possible correlation between precipitation and temperature, wilt symptoms, and pathogen DNA levels in the stalk. The data suggest that LWD impact is most harsh when temperatures climb to 27 °C and above. The disease breaks out slightly at temperatures below 26 °C, with almost no noticeable symptoms (see spring-summer 2018 and autumn 2020).

These results are in line with the data presented by Ortiz-Bustos et al. (2019) [57]. This former work reported that under unfavorable conditions for LWD (low air temperature, 20.9–22.8 °C, and relatively high air humidity, 54.1–58.4%), above-ground symptoms did not appear in the plants despite growth and production variables being altered by the fungus. In contrast, the meteorological conditions identified as optimal for the development of *M. maydis* symptoms were air temperature of 26.2–28.1 °C and humidity of 43.6–45.3%. Hence, the environmental conditions’ influence on disease development is an important aspect with significant implications for the maize growth planning in LWD affected areas. Therefore, the above-accumulated results encourage additional support, which should be provided from future studies. In addition, temperature and humidity are probably not the only factors involved in the disease burst, and other factors yet to be explored should be identified.

Drought stress is indeed a major influencing factor for a plant’s ability to cope with the disease [54]. Over the years of research, it has become evident that soil moisture is one of the most critical factors enhancing late wilt disease progression [54,55,59,60]. *Magnaporthiopsis maydis* DNA levels were nearly 10 times higher in drought-stressed maize sprouts (37 DAS) than in non-stressed plants [27]. The fungus is sensitive to low oxygen conditions in wet soils [55]. In contrast, a high oxygen level promotes the growth of the pathogen’s colonies [48]. Saturated soils result from frequent watering or climate conditions, which reduce late wilt [55]. Indeed, early sowing of maize in Egypt reduced late wilt [59], while late summer planting reduced disease severity in India [60]. This influence mechanism is not entirely clear, and few explanations were suggested regarding the soil conditions and the host plant immunity.

The microorganisms’ communities in the soil that antagonize *M. maydis* may be fixed according to their preference of humidity conditions. Certainly, water availability may be the most central environmental factor affecting the soil’s microbial community and activities [61]. So, floods increase anaerobic conditions that may stimulate lytic organisms to degrade sclerotia and reduce the pathogen’s survival potential.

Regarding the host plant immunity, LWD pathogen infection reduced the number of vascular bundles in the cross-section of the internode [54]. Xylem vessels’ blocking may be the most important factor causing the disease symptoms. Therefore, the values of the phloem area per unit leaf area are crucial. Indeed, these values decreased significantly in drought-stressed and infected plants [54]. In healthy non-stressed maize plants, the high number of vascular bundles in the internodes and the greater phloem area per leaf compared with diseased plants may contribute to a faster translocation rate and LWD resistance.

## 6. Disease Symptoms and Damage

### 6.1. External Symptoms

The LWD first infection signs can be seen in the sprouting phase. *Magnaporthiopsis maydis* can negatively affect the seedlings’ above-ground emergence and cause seedlings’ roots color alternation and necrosis [62]. A seedling assay where different substrates (Perlite, sand, or water agar) were inoculated with the pathogen prior to planting enables the evaluation of LWD’s severe repression effect on the growth of the roots [4,6]. The infected roots showed brown discoloration. In another work [62], small necrotic lesions (2–4 mm long) were observed on the roots of inoculated susceptible plants three weeks after inoculation. Their size increased over time to lengths of 10–14 mm. Interestingly, similar dry, dark-red local lesions near soil level were documented in *M. maydis*-inoculated cotton sprouts (see Section 3) [25].

First-above-ground wilt symptoms appear near the flowering stage (from the R1 silking to the R2 blister stages), approximately 50–60 days after seeding [35]. These first symptoms include rapid wilting of the near-ground leaves that advance upwards during the subsequent two weeks (Figure 7). With the disease progression, the leaves gradually lose their color and become dehydrated. Reddish-brown to yellowish streaks may appear on the lower internode. At that stage, the lower stem dries out (particularly at the internodes) and has a shrunken and hollow appearance. The vascular bundles become occluded, and their color alters from white or light yellow to dark yellow to brownish [6]. Late wilt is commonly associated with infection by secondary pathogens causing the stem symptoms to become more severe [4,63]. Fewer ears are produced, and if kernels are developed, they are often damaged and immature [32] and may be infested with the pathogen. Seed quantity [64] and quality [52] are correlated negatively to disease severity. Eventually, these damaging processes can lead to the plant’s death.

### 6.2. Internal Symptoms

The first symptoms are gradually revealed by flowering (R1-silking, silks visible outside the husks) approximately 50–60 DAS. Later, *M. maydis* typically colonizes the entire stalk, and the vascular tissue is blocked with hyphae and gum-like deposits [35], resulting in water supply suffocation and wilting. Indeed, the leaves of diseased plants have a high content of the amino acid proline, probably associated with water stress due to restricted water flow caused by plugging of the tracheary elements [54]. Infection also resulted in a reduction in the number of vascular bundles in a cross-section of the internode. Under severe conditions, ears can be colonized 12–13 weeks after planting, and the pathogen can colonize the pedicels and move to the pericarp, endosperm, and embryo tissues of the seeds [37]. The process of seed infection also occurs in resistant, apparently non-symptomatic, maize cultivars [32], but at a ca. two-week delay and in lesser amounts of the pathogen (identified by its DNA) [13,32]. Kernel infection can result in seed rot and pre-emergence damping-off [65].

### 6.3. Magnaporthiopsis maydis Crosstalk with the Soil and Plant Microflora

Plants are threatened by a diversity of pathogen species living in complex communities, including also a variety of other non-pathogenic microorganisms. The plant’s microbiome refers to all plant-associated microorganisms, including friendly (beneficial) and hostile (pathogens). The letters can be referred to as the plant’s pathobiome [66]. The plant’s pathobiomes comprise coexisting phytopathogens that affect each other and the plant [67]. They are formed by pathogens inhabiting the same ecological niche and either cooperating or competing for the same plant resources. Two (or more) phytopathogens on the same host can result in significantly different disease outcomes compared to single infections [27]. The natural microorganisms’ communities inhabiting the plant phyllosphere (the above-ground portions of the plant’s habitat) or the rhizosphere (the roots’ surrounding habitat) also include non-pathogenic members that can have protective effects against pathogens.

Reviewing the scientific literature led to the conclusion that notwithstanding *M. maydis*’ pivotal rule, it is not the only cause of late wilt symptoms; instead, it is part of a larger complex of maize pathogenic fungi called the post-flowering stalk rot complex. *Magnaporthiopsis maydis* co-interact with at least two additional partners, *F. verticillioides* and *M. phaseolina* (see [40]). Indeed, *F. verticillioides* is a secondary invader or opportunist developed in late wilt-diseased attenuated maize plants [32,39]. The secondary pathogens rule in LWD severity and can be exemplified in the following case. In the field experiment conducted in the summer of 2018, the *M. maydis*-infested field was suffering from another fungal disease caused by *F. verticillioides* and *Fusarium oxysporum* [41]. This led to more severe dehydration and yield loss. Nevertheless, in the plants that were not affected by the *Fusarium* spp. disease, the drip protection with chemical treatment applied abolished almost wholly any sign of late wilt disease.

To support the synergistic effect between *M. maydis* and *F. verticillioides*, it was shown that those fungi interactions could have a substantial influence on maize plants by an easement of ear rot disease infection [39]. Infection by late wilt pathogen led to increasing systemic infection by *F. verticillioides* from stalks to ears and kernels, attributed to chemical and physiological changes (production of sugars and gums in the xylem of maize stalks). So, the involvement of other pathogens may enhance LWD symptoms and override the chemical protection applied [39]. Still, *M. maydis* interactions with other microorganisms can lead to a beneficial outcome, as proven in other cases [68].

The pathogen’s interactions with maize endophytes (which may play a part in the plant’s resistance factors) have significant implications. Recently [69], 10 fungal species and one bacterial species, *B. subtilis*, endophytes, were isolated from six sweet and fodder maize cultivars with varying susceptibility to late wilt disease (Figure 8). The fungal species belonged to *Trichoderma, Chaetomium, Penicillium, Fusarium, Alternaria* and *Rhizopus* genera. Some are known as phytopathogens (*Fusarium proliferatum* and *Alternaria alternata*), so they play complex roles in their host plant interactions. Two of the endophytes, the fungi *Trichoderma asper**ellum* and *Chaetomium subaffine*, significantly improved the infected plants’ growth indices 42 days after sowing. The fungal species *T. asperellum, Chaetomium cochliodes,*
*Penicillium citrinum* and the bacteria *Bacillus subtilis* treatments were able to reduce the *M. maydis* DNA in the host plant’s roots [69].

In cotton, interactions between *F. oxysporum* (the wilt agent) and *M. maydis* were shown to have an intriguing result [25]. These interspecies relationships are associated with reduced severity of the cotton wilt disease. The infection suppression was maximum when *M. maydis* preceded *F. oxysporum* in the soil compared to when they were inoculated simultaneously. Co-inoculation of the plants with both fungi resulted in some protection compared to *F. oxysporum* alone. In contrast, it exerted little or no protective effect when *M. maydis* was added to the ground after *F. oxysporum* [25]. In a previous study, we reported similar antagonistic relationships between *M. maydis* and *M. phaseolina*, the charcoal rot disease agent in cotton plants [27].

In a plate confrontation assay with *M. phaseolina* and *M. maydis*, the fungi grew toward each other during the six days of incubation in the dark. They formed a clear line between the colonies in the meeting point, and no further progress was possible. No mycoparasitism (growth of one fungus on top of the other) was observed. Moreover, no inhibitory effect by secretion products of one of the pathogens that arrested the other was detected. Additionally, no interlocking of the webs between the two fungi was found [27]. If secreted metabolites are involved in the inhibitory effect, they are effective only at a very short distance (approximately 2 mm).

Such inhibitory active ingredient was recently isolated and identified in the growth medium of *Trichoderma asperellum* (P1) [70]. The 6-Pentyl-α-pyrone compound had strong antifungal activity against *M. maydis*. Nowadays, the mechanism of those specific interactions of *M. maydis* with the other members of the maize pathobiome is still obscure. Yet, we can infer the mechanism from other plant-related fungi (see a summarization in [39]), which can provide us with a lead for follow-up works.

## 7. Diagnostics Techniques

A variety of experimental and lab protocols developed over the years to study *M. maydis* and LWD progression, assess potential control strategies and identify key intervention points, which allow us to maximize their impact. Those methods are the topic of a recent review published by us [13]. So, we will cover this topic here in brief. The first research stages in studying the disease usually focus on *M. maydis* isolation from infested soils and plant tissues using media plates and identifying the pathogen using taxonomic keys. Traditional identification methods of the pathogen include microscopic and colony morphology characterization. The latter include spore development and release and sclerotium body maturation. Any information in this regard is important because the microscopic phenotypes of *M. maydis* described in the literature are limited (the few examples include [4,13,29,32,71]). The molecular diagnostic techniques (PCR and qPCR) are a sensitive and precise method for the final confirmation of pathogen identity.

Additionally, spore germination and the detached root assays are essential techniques for studying *M. maydis* behavior under challenging environmental conditions, as demonstrated previously [48,72]. Seed germination, sprout growth and maturation and cob production are key phenological stages for in vivo evaluation of crucial pathogenicity aspects. The pathogen impact on the maize host can be assessed by measuring its ability to invade the inner tissues of seeds [37], which influence their germination rate [71], inhibit the seedlings thrived [6], and their establishment in the vascular tissues, and eventually disrupt normal growth and cob production. These stages can be studied individually, or some could be used together to evaluate a particular intervention treatment, as previously demonstrated [42].

It is important to realize that a scientific plan focused on the pathogenesis’ study or developing disease control cannot be established entirely based on field experiments during the growing season. This is due to the high demand for time, resources, and labor. Moreover, the lengthy period until results are received, and the variations in environmental settings lead to inconsistent results. Therefore, other relatively fast assays should be considered. For instance, to rapidly map the *M. maydis* population and identify pathogenic variations within the population, the fungal culture filtrate seed germination assay is a rapid and efficient method, as previously reported [13,71].

Additional essential methods are available for studying *M. maydis* virulence behavior and host resistance. These methods include spore suspension injection into the lower stalk of maize breeding/germplasm lines to assess their responses to late wilt [29,40,73]. Also, chemical and histological approaches are powerful tools to identify differences between susceptible and resistant maize hybrids [74]. Various techniques were developed for the same purpose, which includes: examining maize root exudates embedded media that influence fungal radial growth [71]; tracking the appearance of necrotic lesions on the roots [25,62]; monitoring *M. maydis* infection by measuring canopy temperature and crop water stress index [9]; injecting *M. maydis* culture filtrate (secreted metabolites) into LWD-sensitive mature maize plants; evaluating stem tissue color alternation and water conductivity to scale the pathogen isolates’ aggressiveness [71].

## 8. Control Strategies

Over the years, various control methods were suggested to cope with LWD, some of which gained positive results in reducing the disease in commercial fields. Reviewing these methods with their pros and cons merit an independent paper. Therefore, this topic will be covered here briefly. The different LWD prevention methods that have been examined include balanced soil fertility [55,75], watering the field [56], soil solarization [76], and allelochemical [62] and chemical options [77,78]. In 2017–2018, Israel’s search for a chemical application to control LWD led to an economically feasible solution [41,47,52]. The successful preventive protocol is based on changing the maize cultivation method to pairs of adjacent rows using a dripline irrigation system and the sophisticated integration of Azoxystrobin-based pesticide mixtures in a three-interval schedule adapted to key points in the disease development. Unfortunately, the rapid development of resistance to this fungicide and the consequential control failure in other crops [79] indicate that future challenges lie ahead and that other options should be explored.

Hence, there is a growing trend of replacing pesticide use with more environmentally friendly solutions [80] that reduce the environment’s impact and improve human health safety. Indeed, green biological approaches to control *M. maydis* and other phytopathogens are at the forefront of the current scientific global effort. Two eco-friendly strategies to restrict late wilt are currently in this scientific focus.

First, the use of *Trichoderma* sp. and other beneficial microorganisms as a biocontrol agent has been demonstrated in the past with promising potential (most recently [81,82]). Similarly, we conducted three years of research with new *Trichoderma* species and identified highly potential LWD biocontrol candidates. These include *T. longibrachiatum* (T.7407 from marine source [83]), *T. asperelloides* (T.203) and *T. asperellum* (P1), an endophyte isolated from maize seeds [69]. These isolates successfully prevented the pathogen’s growth in culture plates, reduced its establishment and development in the plants, and improved growth and crop indices under field conditions at the season end [58,84].

Second, soil conservation practices that promote and maintain *M. maydis* and antagonize soil mycorrhizal fungi proved to be efficient (summarized by [81]). Research carried out in other phytoparasitic fungi suggested that under no-till cropping and crops in a rotation or cover crops could help build mycorrhizal communities shared by several crops [85]. Adjusted tillage systems and cover crops to maintain the integrity and continuity of the soil mycorrhizal networks may improve the plants’ resistance to soil diseases [86], including LWD [87]. Recently it was shown for the first time that the wheat-corn rotation without tillage resulted in a significant increase in plant growth and yield compared to the clover-corn growth cycle and the control [88]. Moreover, a 67% improvement in plant health and a 96% decrease in pathogen infection was achieved in the no-tillage wheat-corn rotation treatment compared to the control. These findings are a first step towards developing eco-friendly biological protection against the late wilt agent adapted to sustainable agriculture that may reduce other maize diseases.

Regardless of this research progress, none of these control methods are currently being used in Israel. Instead, LWD is managed economically in most countries by developing genetically resistant maize cultivars (Figure 9) [73,74,89,90]. The Agricultural Research Center in Giza, Egypt, operates a national maize program to identify LWD resistance sources. Since 1980, the release of such resistant genotypes has significantly reduced the economic impact of late wilt in Egypt [59]. A program for identifying resistant cultivars has been active in Israel for nearly two decades. Yet, the use of resistant maize varieties is not without its problems. *Magnaporthiopsis maydis* can spread in non-symptomatic resistant plants and infect their seeds [32,50]. Also, highly aggressive isolates of *M. maydis* [9,14] may threaten these resistant maize cultivars. Indeed, the growth of a resistance cultivar for many years in the exact location may result in gradual LWD immunity weakening [13,47].

## 9. Future Perspectives

The late wilt of maize is considered endemic to several countries. It has been reported officially in only seven countries, and there are unconfirmed reports of the disease in three additional countries (Figure 1). For this reason, the study of LWD is progressing relatively slowly and remains far behind the research forefront of more familiar plant diseases. Most of the work on this pathogen has focused on means to restrict its distribution, control its damage, and improve our ability to detect and monitor it. Hence, the essential information regarding the biology of the pathogen and its interactions with the host plant remains obscure. Even the genome of this fungal species has never been sequenced. Such vital information, yet to be obtained, includes the potential toxins it may produce, the pathogen enzymatic array secreted during pathogenesis, and the cellular network of signal transduction pathways that coordinated and synchronized these events.

Other fungal toolkits that involve the host’s penetration, establishment, and activity disruption are poorly understood. For example, during the fungal attack on the surface of the roots, *M. maydis* produce appressorium-like structures by curling and twisting side branches that arise from the primary elongating hyphae [35]. These structures’ formation and function were described but should be studied in a more in-depth fashion. Also, during the last stage of the disease, the maize vessels become blocked with hyphae and dark gum-like substances [35,36] that should be identified. In this context, the laccase enzyme studied recently [51] exemplifies the importance of such a database for future coping with this emerging threat.

Another vital issue is mapping the *M. maydis* population and identifying pathogenic variations within the pathogen population and the relationship between fungal varieties. Studies targeting this aim were carried out in Egypt [14] and Spain [91], but are needed in other LWD distribution areas. Such efforts may improve our understanding of this pathogen and enable the establishment of feasible future control practices. Indeed, it was shown that different competitive abilities exist within the *M. maydis* population in Egypt and that pathogenic diversity in Spain enabled determining the levels of aggressiveness of this pathogen. Such research data play a crucial role in decision-making and risk assessment before starting a commercial maize field growth season. There are also substantial knowledge gaps regarding the interactions of *M. maydis* with its physical and biological surroundings. For example, possible additional alternative plant hosts should be tested. Such new species could enhance the pathogen’s spread and survival.

Finally, new control practices that could minimize disease or contain the pathogen spread need to be developed, and sources of genetic resistance in maize must be sought. Chemical control of LWD is a strong intervention step to protect susceptible maize cultivars against LWD in highly infected areas [41,42,52]. Still, the rapid development of resistance to fungicides [92] and their environmental and health potential risk encourage green solutions [80]. While developing environmental-friendly options is the focus of many novel research works, there is still a need for improved protocols, as demonstrated by Elshahawy and El-Sayed, 2018 [82]. These researchers suggested maximizing the efficacy of Trichoderma to control *M. maydis* using freshwater microalgae extracts.

Eco-friendly and cost-effective genetic intervention is the preferred option to mitigate LWD losses [93,94]. Precede to genetic intervention, stable sources of resistance to *M maydis* are essential in maize breeding. Yet, little information is available regarding resistance to LWD alongside high yielding in maize [95]. The inheritance of resistance to LWD is complex with significant genotype × environment interaction. Hence, direct selection for LWD resistance is likely to be less effective [90]. Nevertheless, DNA markers could be used as an adequate substitute for such traits in maize, for which apriority identification and validation of closely linked markers is essential. Quantitative Trait Locus/Loci (QTL) conferring LWD tolerance have been detected and validated as a step towards this end [89,90]. Several studies indicate the possibility that many genes are involved in controlling LWD resistance in maize [96]. Further research investigation is essential to identify stable QTL with large phenotypic variation explained [90].

## Figures and Tables

**Figure 1 jof-07-00989-f001:**
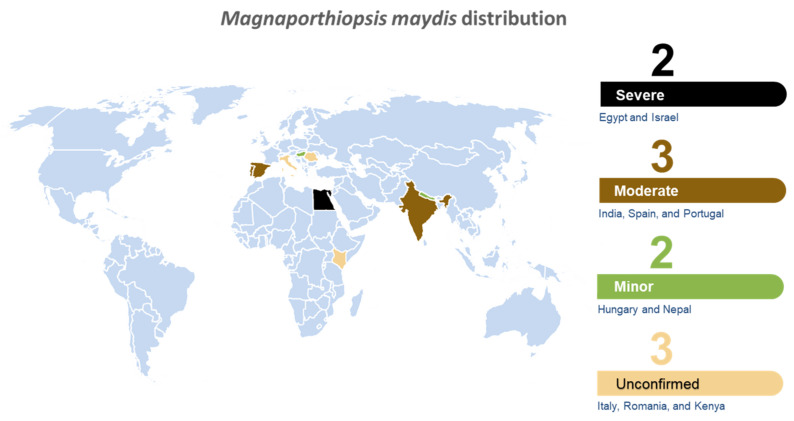
World distribution map for *Magnaporthiopsis maydis*. Disease severity is appraised according to the literature reports and is based on three categories: severe (4, Egypt and Israel); moderate (3, India, Spain and Portugal); minor (2, Hungary and Nepal); and not certain/unconfirmed reports (1, Italy, Romania and Kenya).

**Figure 2 jof-07-00989-f002:**
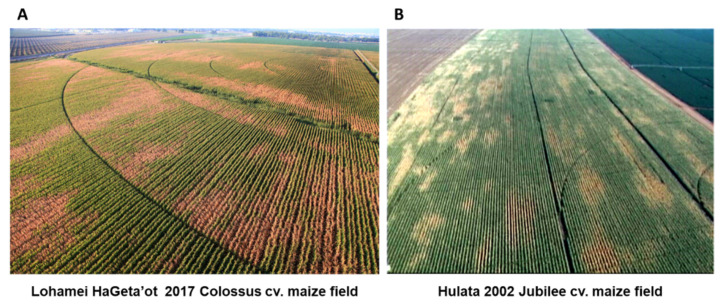
Aerial photographs of maize late wilt diseased commercial fields (photographed by Asaf Solomon). Both fields are located in northern Israel. (**A**) Photo from 2017 of an area situated near Lohamei HaGeta’ot. The sensitive fodder maize (*Zea mays* L.) cultivar is Colossus from HSR Seeds, Orbost, Australia, supplied by CTS, Hod Hasharon, Israel. (**B**) Photo from 2017 of an area situated near Hulata. The susceptible sweet maize cultivar is Jubilee from SRS Snowy River Seeds, Australia, supplied by Green 2000 Ltd., Israel. The brown areas are wilting plants infected by *Magnaporthiopsis maydis*.

**Figure 3 jof-07-00989-f003:**
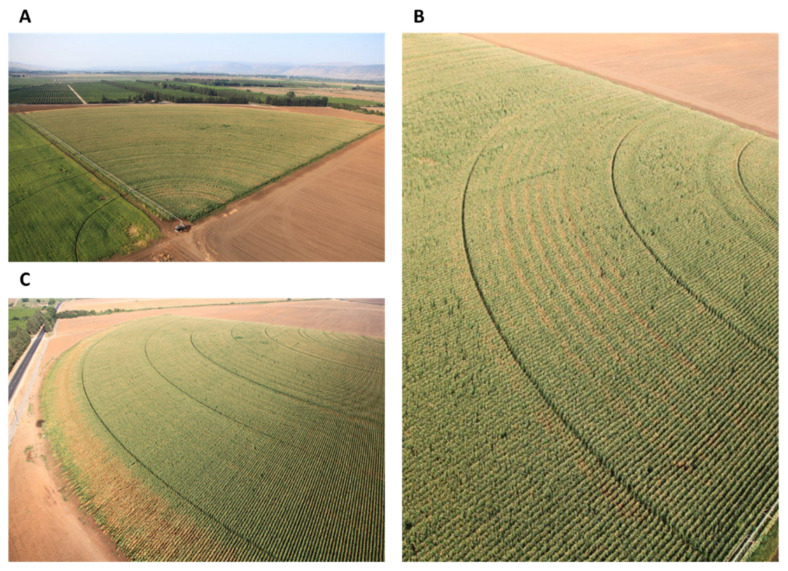
Aerial photographs of a maize field near Kfar Blum (Hula Valley, Upper Galilee, northern Israel). The photos were taken by Asaf Solomon on 3 September 2012. (**A**) The whole field. A circular irrigation system watered the field. In some areas, ununiform irrigation, such as blockage sprinklers (**B**) or limited water supply at the field edge (**C**), evoked the late wilt disease burst.

**Figure 4 jof-07-00989-f004:**
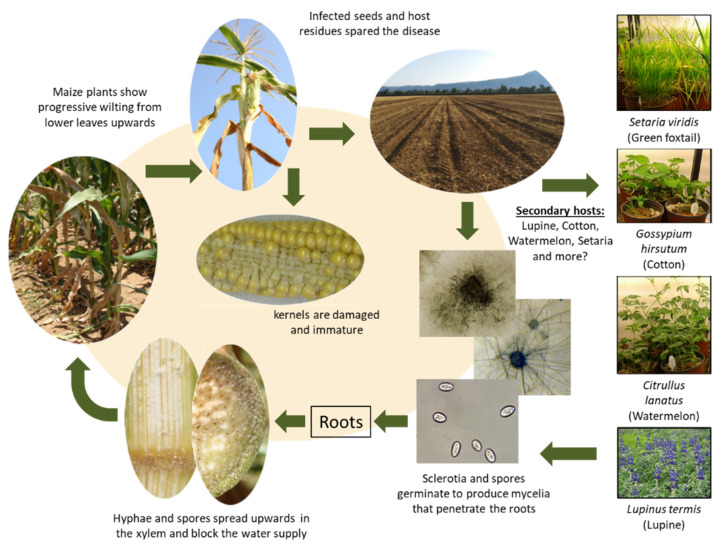
Disease cycle of *Magnaporthiopsis maydis*.

**Figure 5 jof-07-00989-f005:**
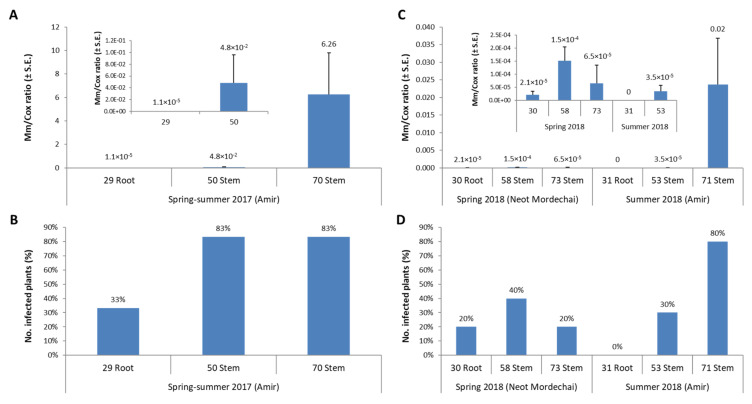
qPCR diagnosis of late wilt pathogenesis in light and severe disease outbreak. Molecular qPCR-based tracking of the *Magnaporthiopsis maydis* DNA in the 2017 field experiment (intense disease outburst, (**A**,**B**)) [52] compared to the spring (slight disease) and summer (severe disease) 2018 field experiments’ results (**C**,**D**) [41]. All experiments were performed at the same site. The Y-axis (in (**A**,**C**)) indicates *Magnaporthiopsis maydis* proportionate DNA abundance normalized to the cytochrome c oxidase (*COX*) DNA. Bars represent a mean of 10 replicates. Error bars show standard errors.

**Figure 6 jof-07-00989-f006:**
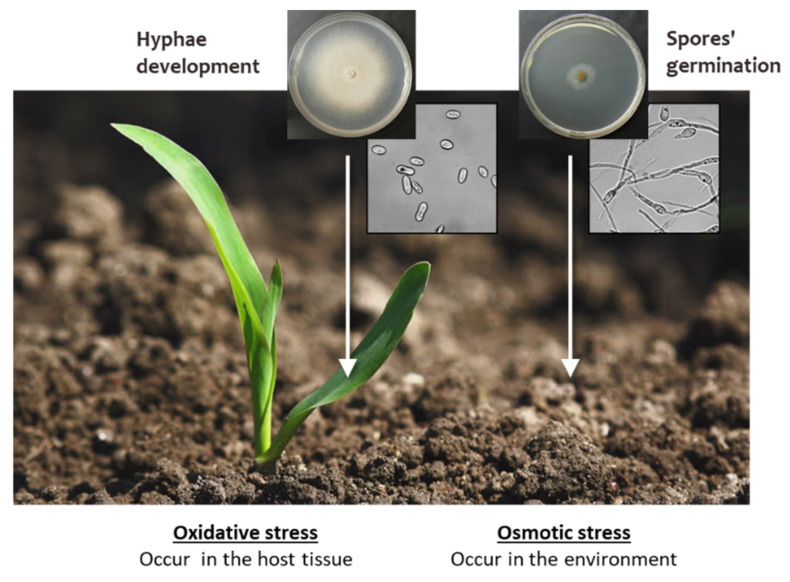
Environmental pressures on spores’ germination and hyphal growth led to different growth responses. The pathogen’s spore germination in the soil and hyphal development inside the host tissues react differently to oxidative and salt pressures [48]. It is assumed that hyphae developed resistance to oxidative stress. Such stresses were induced as a default plant response to the pathogenic attack. This tolerance enables *Magnaporthiopsis maydis* to invade and establish within the host during the growth period. Still, the ability of hyphae to encounter osmotic and ionic stresses that are common in the soil is limited. In contrast, conidia are repressed by a high oxidative environment and more easily germinated in a salty environment, their natural surroundings.

**Figure 7 jof-07-00989-f007:**
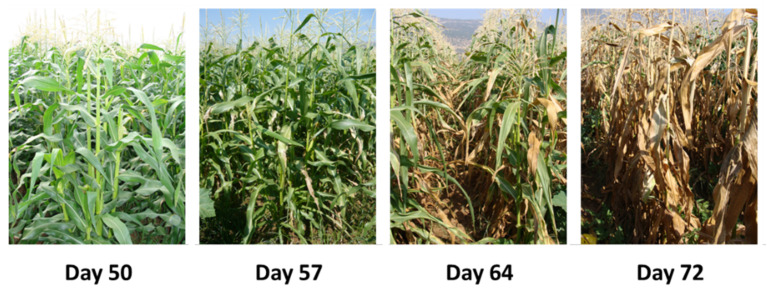
Disease progression in maize late wilt-susceptible Jubilee cv. plants from 50 to 72 days after sowing. The photos were taken during a field experiment conducted in 2008 for assessing cultivar resistance to *Magnaporthiopsis maydis* in an infected sweet corn field in the Hula Valley (Upper Galilee, northern Israel) [32].

**Figure 8 jof-07-00989-f008:**
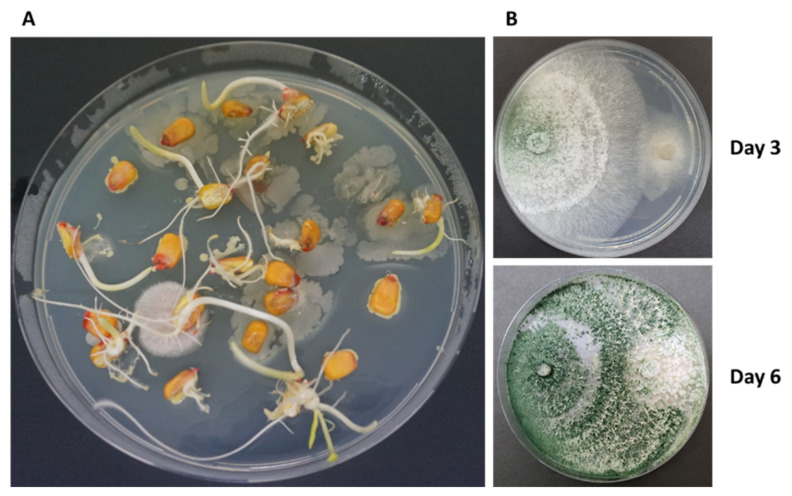
Maize seeds microflora. (**A**) Maize grains were disinfected externally, cut lengthwise, and placed on a potato dextrose agar (PDA) medium with the cutting surface turned downwards. Petri dishes were maintained in the dark at 28 ± 1 °C for 2–3 days. (**B**) Plate mycoparasitism assays—*T*. * Asperellum* (P1, **left** side) vs. *Magnaporthiopsis maydis* (**right** side) [69].

**Figure 9 jof-07-00989-f009:**
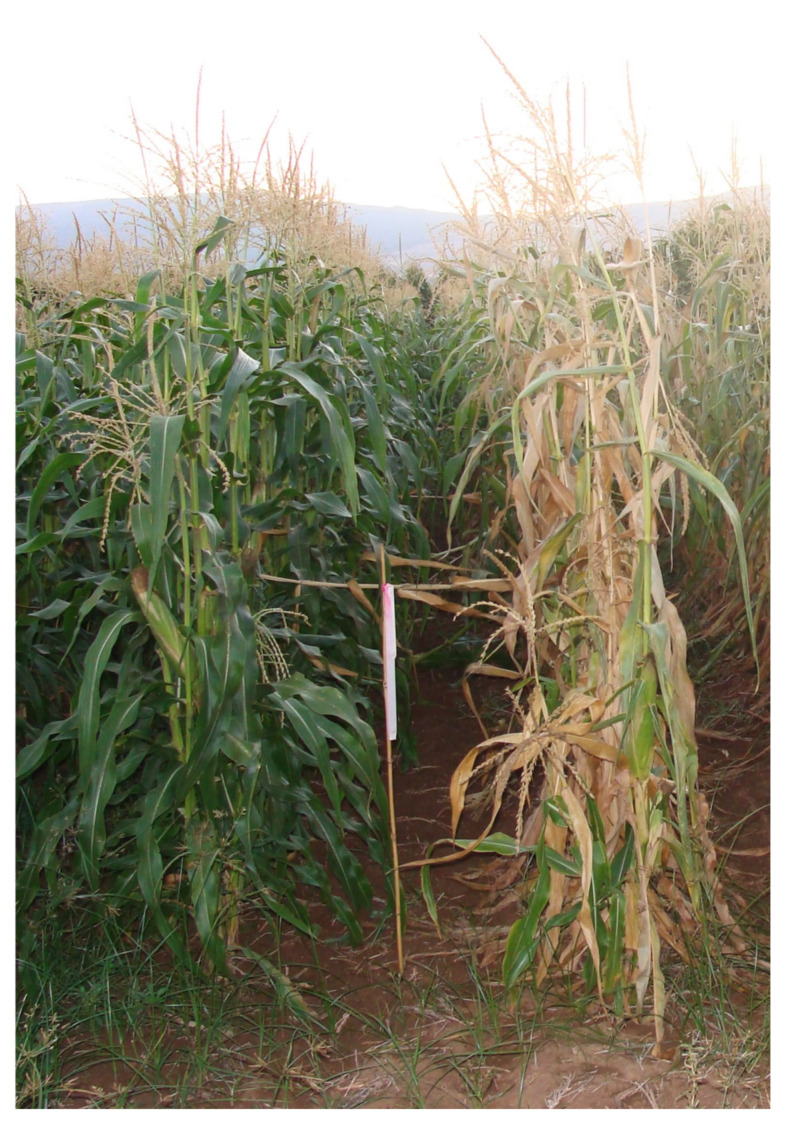
Field assay for assessing sweet maize cultivars’ resistance to late wilt disease. The experiment was conducted in 2009 in a commercial field located near Kibbutz Hulata (Hula Valley, Upper Galilee, northern Israel). The photo was taken 75 days after sowing (ca. 20 days after fertilization). On the left, the resistant cultivar Royalty. On the right, the susceptible maize hybrid, Jubilee, shows severe dehydration symptoms.

**Table 1 jof-07-00989-t001:** Environmental conditions, maize late wilt disease symptoms, and *Magnaporthiopsis maydis* infection levels in the growing seasons of 2016–2020 in northern Israel ^1^.

Location and Year	Dates	Average Temp.	Precipitation	Leaf and Stem Tissue Showing Wilting	*M. maydis* DNA	Reference
Amir 2016 (spring-summer)	25/5–2/8 (75 DAS)	27 °C	0.6 mm	60% (69 DAS) 100% (75 DAS)	0.05 (60 DAS) 7.8 × 10^−5^ (75 DAS)	[42]
Amir 2017 (spring-summer)	24/5–2/8 (70 DAS)	27 °C	0 mm	73%	6.26	[52]
Neot Mordechai 2018 (spring-Summer)	23/4–5/7 (73 DAS)	25 °C	30 mm	Less than 10%	6.5 × 10^−5^	[41]
Amir 2018 (summer)	21/6–5/9 (71 DAS)	28 °C	3 mm	72%	0.02	[41]
Gadash Farm 2019 (summer)	6/8–30/10 (85 DAS)	26 °C	53 mm	30%	5.3 × 10^−4^	[58]
Gadash Farm 2020 (autumn)	10/9–1/12 (82 DAS)	23 °C	141 mm	Less than 10%	4.5 × 10^−4^	[58]

^1^ The experiments were conducted in maize fields in the Hula Valley in Upper Galilee, northern Israel, located about 10 km from each other. The late wilt susceptible sweet maize Prelude cv. was assessed in all seasons. Average meteorological data were according to Israel Northern Research and Development Meteorological Station data—Hava 1. Dehydration percentage—the plants were classified as wilted when wilt symptoms appeared on the cobs’ spathes. *M. maydis* DNA—lower stem qPCR results of specific *Magnaporthiopsis maydis* DNA fragment normalized to the cytochrome c oxidase (*COX*).

## Data Availability

The data presented in this study are available on request from the corresponding author.

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
