# Peer review of "A Review: Late Wilt of Maize—The Pathogen, the Disease, Current Status, and Future Perspective"

_jof, 2021, doi:10.3390/jof7110989_

Round 1

Reviewer 1 Report

See comments file and revised file of the manuscript

Author Response

Responses to Reviewer 1’s comments

We thank the reviewer for investing substantial efforts, which is contributing significantly to this manuscript. The remarks and suggestions improved this scientific paper remarkably and made it more accurate, clear, focused, and well-structured. Your contribution is greatly appreciated.

General comments:

The manuscript presents a very valuable and complete review on the fungal pathogen causing late wilt of maize, the disease, the literature, and outcomes on the topic and future prospects. From that point of view, the manuscript is interesting, is well-written, and falls within the journal scope. However, I found some flaws and issues the author should address in order to consider the acceptance of the manuscript for publication.

Thank you for the constructive and essential comments. We accepted all your comments and suggestions, as detailed in the following point-by-point response.

Names of pathogens must be stated in full (meaning the full genus name) the first time they are mentioned and also always at the beginning of a sentence, as well as in figures’ captions and tables’ headings. This applies to Magnaporthiopsis maydis. Please check line 61 and below.

The reviewer is right. We carefully double-checked the entire manuscript and verified that all fungi names were correctly stated. Indeed this subject was corrected in several places.

The author considers DNA presence in maize tissues equal to disease. However, he must be cautious about this assumption. It is frequent that even the fungus being in the inner tissues, the disease does not develops, what happens when environmental conditions do not favour disease outbreaks (see lines 319-333) or in genetically tolerant maize (see lines 593 and 594). Besides, there is no known correlation between DNA amount and disease severity or yield losses. In other words: we cannot know how much DNA is needed to reach, for instance, 50% symptoms severity or reducing the yield by 50%. I would recommend the author to include a sentence stressing that disease severity and yield loss cannot be inferred from neither the presence nor the amount of DNA in tissues.

We agree with this remark. The following sentence is incorporated into the text (lines 307-309): “While that has been said, it should be emphasized that disease severity and yield loss cannot be inferred from the presence or amount of DNA in tissues. Such correlation should be examined and established in follow-up studies.”

A third point is that the information presented in sections 5.3 to 5.5 results somehow messy and blurred. I suggest merging sections 5.3. (Environmental conditions favorable for the pathogen’s thriving), 5.4. (Environmental pressures’ differential effect on spores and hyphal growth) and 5.5. (Water deficiency and late wilt disease) and then splitting the information into:

5.3 Effects of abiotic factors on the biology of Magnaporthiopsis maydis

5.2 Environmental conditions and disease development. In 5.2, favourable conditions for the disease can be developed, making particular emphasis on the influence of water deficiency

This is good advice. Thank you. The paragraphs were re-ordered according to the reviewer’s recommendation. Some minor text changes were made to maintain the text continuity and clarity.

Other comments:

Section 5.3. Concerning the influence of environmental conditions on disease development (lines 319-333), it is stated that future studies should investigate the impact of temperature on disease development. However, Ortiz-Bustos et al. (2019) already reported temperature and humidity conditions that were favourable (and unfavourable) for disease development (Plant Pathology 68(8): 1555–1564. DOI: https://doi.org/10.1111/ppa.13070). Also in this work the authors found that, under unfavourable conditions for disease (low air temperature and

relatively high air humidity), above-ground symptoms did not appear in the plants despite growth and production variables being clearly altered by the fungus. I would recommend the author to consider this work with regard to the influences of environmental conditions and irrigation on late wilt and maize yield.

This is important advice. We added the information to the text and improved the interpretation regarding this topic (lines 380-391):

“These results are in line with the data presented by Ortiz-Bustos et al. (2019) [58]. This former work reported that under unfavorable conditions for LWD (low air temperature, 20.9-22.8 ºC, and relatively high air humidity, 54.1-58.4%), above-ground symptoms did not appear in the plants despite growth and production variables being altered by the fungus. In contrast, the meteorological conditions identified as optimal for the development of M. maydis symptoms were air temperature of 26.2-28.1 ºC and humidity of 43.6-45.3%. Hence, the environmental conditions’ influence on disease development is an important aspect with significant implications for the maize growth planning in LWD affected areas. Therefore, the above-accumulated results encourage additional support, which should be provided from future studies. In addition, temperature and humidity are probably not the only factors involved in the disease burst, and other factors yet to be explored should be identified”.

References. There are several references that are mentioned twice in the References section. I highlighted them in the pdf file.

We corrected the duplicate references.

Some other minor points are indicated in the revised pdf file.  

All minor points suggested were accepted and corrected according to the reviewer’s advice. See the point-by-point response to each remark in the reviewer response PDF file.

Reviewer 2 Report

The review paper (A Review: Late Wilt of Maize—The Pathogen, the Disease, Cur3 rent Status and Future Perspective) is very interesting. The paper provides essential information about LWD, pathogen spread, disease implications and control methods. The following points should be improved before publication:

In the introduction, please write about the importance of the disease in one paragraph. It is better to transfer the last sentence in the first paragraph to the beginning in the second paragraph.

You wrote (The disease was first identified and reported in Egypt in 1961-1962 and gradually reported in other countries). The importance of the disease in the second paragraph should be started with this sentence then move to the importance of disease in Egypt and Israel and go through the importance of LWD in the US.

The aim of the study should be started with (The variety of tools developed to study and monitor M. maydis were recently reviewed by us and will be summarized here briefly) and then finish with the current review ………etc.

In M. maydis crosstalk with the soil and plant microflora, this point is very interesting for the readers especially with other pathogens such as F. verticillioides and M. phaseolina. I would suggest you add more explanation about the mechanism of effect toward each others.

You wrote (In cotton, interactions between Fusarium oxysporum (the wilt agent) and M. maydis were shown to have an intriguing result). This paragraph should be transferred to be with the effect of M. maydis on other pathogens as one part and the second part as you explained the relationship between M. maydis and other endophytes.

Line 500 F. oxysporum should be italic.

In the future perspectives, the following sentence (Such vital information, yet to be obtained, includes the potential toxins it may produce, the pathogen enzymatic array secreted during pathogenesis, and the cellular network of signal transduction pathways that coordinated and synchronized these events) should be transferred with the previous paragraph.

I think still you can add more points to the future perspectives such as the mechanism of disease resistance and control strategies. Please try to improve the future perspectives.

Author Response

Responses to Reviewer 2’s comments

We would like to express our sincere appreciation to the reviewer for the important and helpful suggestions and advice. The time and effort invested are greatly appreciated, and without a doubt, contributed to the manuscript and significantly improved it. Thank you.

comments:

The review paper (A Review: Late Wilt of Maize—The Pathogen, the Disease, Current Status and Future Perspective) is very interesting. The paper provides essential information about LWD, pathogen spread, disease implications and control methods. The following points should be improved before publication:

Thank you for the constructive and essential comments. We accepted all your comments and suggestions, as detailed in the following point-by-point response.

In the introduction, please write about the importance of the disease in one paragraph. It is better to transfer the last sentence in the first paragraph to the beginning in the second paragraph.

The sentence was transferred from the first paragraph to the second paragraph, as suggested.

You wrote (The disease was first identified and reported in Egypt in 1961-1962 and gradually reported in other countries). The importance of the disease in the second paragraph should be started with this sentence then move to the importance of disease in Egypt and Israel and go through the importance of LWD in the US.

The second paragraph was re-ordered as suggested. It is indeed now more coherent and focused. Thank you.

The aim of the study should be started with (The variety of tools developed to study and monitor M. maydis were recently reviewed by us and will be summarized here briefly) and then finish with the current review ………etc.

The paragraph was re-edited and improved and is now reads:

“The variety of tools developed to study and monitor M. maydis were recently reviewed by us [14] and will be summarized here briefly. Also, reviewing the global efforts to develop LWD control methods is worthy of a separate paper and will be presented here in summary. The current review summarizes the knowledge accumulated on LWD, its causal agent, pathogen spread, disease implications, and control methods. Another essential aspect discussed is a future perspective on risks and knowledge gaps that should be addressed.“ (lines 56-62)

In M. maydis crosstalk with the soil and plant microflora, this point is very interesting for the readers especially with other pathogens such as F. verticillioides and M. phaseolina. I would suggest you add more explanation about the mechanism of effect toward each others..

Thank you for this remark. Unfortunately, the mechanism of those specific interactions with M. maydis is still obscure. Yet, we can infer the mechanism from our results and other plant pathogenic fungi, and this can provide us with a lead for follow-up works.

The following explanations were added to the text

Lines 495-501: “To support the synergistic effect between M. maydis and F. verticillioides, it was shown that those fungi interactions could have a substantial influence on maize plants by an easement of ear rot disease infection [39]. Infection by late wilt pathogen led to increasing systemic infection by F. verticillioides from stalks to ears and kernels, attributed to chemical and physiological changes (production of sugars and gums in the xylem of maize stalks). So, the involvement of other pathogens may enhance LWD symptoms and override the chemical protection applied [39]”.

Lines 523-536: “In a plate confrontation assay with M. phaseolina and M. maydis, the fungi grew toward each other during the six days of incubation in the dark. They formed a clear line between the colonies in the meeting point, and no further progress was possible. No mycoparasitism (growth of one fungus on top of the other) was observed. Moreover, no inhibitory effect by secretion products of one of the pathogens that arrested the other was detected. Additionally, no interlocking of the webs between the two fungi was found [27]. If secreted metabolites are involved in the inhibitory effect, they are effective only at a very short distance (approximately 2 mm).

Such inhibitory active ingredient was recently isolated and identified in the growth medium of Trichoderma asperellum (P1) [70]. The 6-Pentyl-α-pyrone compound had strong antifungal activity against M. maydis. Nowadays, the mechanism of those specific interactions of M. maydis with the other members of the maize pathobiome is still obscure. Yet, we can infer the mechanism from other plant-related fungi, which can provide us with a lead for follow-up works.”

You wrote (In cotton, interactions between Fusarium oxysporum (the wilt agent) and M. maydis were shown to have an intriguing result). This paragraph should be transferred to be with the effect of M. maydis on other pathogens as one part and the second part as you explained the relationship between M. maydis and other endophytes.

Since Fusarium oxysporum is a pathogen and not an endophyte, we think it is better to present these results in a separate paragraph that will discuss pathogen-pathogen interactions.

Line 500 F. oxysporum should be italic.

Corrected as advised.

In the future perspectives, the following sentence (Such vital information, yet to be obtained, includes the potential toxins it may produce, the pathogen enzymatic array secreted during pathogenesis, and the cellular network of signal transduction pathways that coordinated and synchronized these events) should be transferred with the previous paragraph.

Corrected as per the reviewer’s advice.

I think still you can add more points to the future perspectives, such as the mechanism of disease resistance and control strategies. Please try to improve the future perspectives.

As suggested by the reviewer, the future perspectives section was reworked and improved. The following new paragraphs were added (lines 675-695):

“Finally, new control practices that could minimize disease or contain the pathogen spread need to be developed, and sources of genetic resistance in maize must be sought. Chemical control of LWD is a strong intervention step to protect susceptible maize cultivars against LWD in highly infected areas [41,42,53]. Sill, the rapid development of resistance to fungicides [92] and their environmental and health potential risk encourage green solutions [80]. While developing environmental-friendly options is the focus of many novel research works, there is still a need for improved protocols, as demonstrated by Elshahawy and El-Sayed, 2018 [82]. These researchers suggested maximizing the efficacy of Trichoderma to control M. maydis using freshwater microalgae extracts.

Eco-friendly and cost-effective genetic intervention is the preferred option to mitigate LWD losses [93,94]. Precede to genetic intervention, stable sources of resistance to M maydis are essential in maize breeding. Yet, little information is available regarding resistance to LWD alongside high yielding in maize [95]. The inheritance of resistance to LWD is complex with significant genotype × environment interaction. Hence, direct selection for LWD resistance is likely to be less effective [90]. Nevertheless, DNA markers could be used as an adequate substitute for such traits in maize, for which apriority identification and validation of closely linked markers is essential. Quantitative Trait Locus/Loci (QTL) conferring LWD tolerance have been detected and validated as a step towards this end [89,90]. Several studies indicate the possibility that many genes are involved in controlling LWD resistance in maize [96]. Further research investigation is essential to identify stable QTL with large phenotypic variation explained [90]”.